# The Influence of Local Microstructure Inhomogeneities on Local Drying Kinetics during Freeze-Drying

**DOI:** 10.3390/pharmaceutics14102132

**Published:** 2022-10-07

**Authors:** Sebastian Gruber, Maximilian Thomik, Nicole Vorhauer-Huget, Lukas Hans, Evangelos Tsotsas, Petra Foerst

**Affiliations:** 1Chair of Process Systems Engineering, TUM School of Life Sciences, Technical University of Munich, Gregor-Mendel-Str. 4, 85354 Freising, Germany; 2Food Process Engineering, TUM School of Life Sciences, Technical University of Munich, Weihenstephaner Berg 1, 85354 Freising, Germany; 3Department of Thermal Process Engineering, Institute of Process Engineering, Otto-von-Guericke University Magdeburg, Universitaetsplatz 2, 39106 Magdeburg, Germany

**Keywords:** freeze-drying, microstructure, drying kinetics, lyomicroscope, sucrose, maltodextrin, image analysis

## Abstract

Freeze-drying is a gentle drying technique to dry high value products, such as pharmaceuticals, without impacting the quality of the product. However, this method is very time and cost intensive. It is known that larger pores reduce the duration of primary drying due to facilitated mass transport. However, next to the pore size, other structural parameters exist whose influence on drying kinetics is still unknown. Therefore, the aim of this article is to investigate the influence of the microstructure (pore size, shape and orientation) on local primary drying kinetics. In the study, freeze-drying experiments on maltodextrin and sucrose solutions (c_1_ = 0.05 and c_2_ = 0.15 w/w) were carried out in a lyomicroscope. Two-dimensional images were recorded during the whole drying process and in the dry state and analyzed on the movement of the sublimation front, pore size, orientation and shape. Different microstructures were created by using different freezing parameters, namely two different cooling rates and solid concentrations. It could be shown that for pores with a high aspect ratio, the pore orientation was more important for the drying kinetics than the pore size, while for pores with a lower aspect ratio the pore size was the decisive parameter.

## 1. Introduction

The freeze-drying process, or better known as lyophilization, is a typical drying technique in the pharmaceutical industry to preserve products such as biopharmaceuticals or vaccines. The process can be divided into three steps: 1. Freezing, 2. Primary drying and 3. Secondary drying [1,2,3,4]. The freezing step is the most important since it defines the morphological structure of the product and thus influences primary and secondary drying [5,6]. In the primary drying phase, the ice is removed by sublimation at low pressure and temperature. In the secondary drying step, the bound water is removed by desorption [1]. Freeze-drying of pharmaceuticals typically takes place in vials, which are placed on heated shelfs inside the freeze-dryer [7]. Due to the stochastic event of the freezing-process, the morphological structure between vials tends to be quite heterogeneous. This, however, will lead to different drying behaviors and thus can lead in the worst case to product damage or poor product quality [1,8]. In addition to product quality, the economy of the process is also important. For that, process time and energy costs are vital. To shorten drying time, the shelf temperature can be increased [9]; however, critical parameters such as glass transition temperature of the maximum freeze concentrated solution (Tg’) or collapse temperature (Tc) need to be taken into account [1,2,10,11,12]. Pikal 2007 [1] demonstrated that increasing the temperature by 5 °C leads to half the drying time. A safer way to shorten drying time is by controlling pore size distribution. Here, it could already be demonstrated that larger pores lead to faster primary drying because of the lower water vapor transport resistance [3,4,13]. To characterize the microstructure of a freeze-dried sample, pore size distribution is usually calculated. Nevertheless, there are also other parameters to describe the microstructure in detail, such as pore shape and pore orientation. However, in the literature there are only a few studies that describe the microstructure of a freeze-dried sample by such parameters. To characterize the freeze-drying process in more detail, multiple methods exist [14,15,16,17]. Usually, these methods focus on the global sublimation rate and can be measured by local temperature measurements or gravimetric methods, for example [18,19]. However, the relationship between microstructure and drying kinetics remains unclear. To understand the freeze-drying behavior in more detail, combined information on drying kinetics and microstructure is necessary. To analyze microstructural properties, different imaging methods exists. They are usually classified into 2D and 3D imaging methods. For 2D imaging, light microscopy is typically used, whereas for 3D imaging, X-rays or neutrons are used [15].

Foerst et al. 2019 [3] used X-ray tomography to analyze the freeze-dried cake. They used the tomography data to analyze mass diffusivity by using a simulation tool. They could demonstrate that annealing leads to larger pores and thus to a faster drying time [3].

Siebert et al. 2019 [20] investigated the influence of different drying techniques on the microstructure of carrot discs. They compared freeze-drying, microwave freeze-drying and microwave vacuum drying. They found out that the carrot discs dried rather radial. The findings were supported by the pore structure of the discs. While the outer part was quite dense and collapsed, the inner central part remained porous.

Gruber et al. 2021 [15] used X-ray tomography to study the freeze-drying process in situ. Here they showed that this method is a valuable tool to analyze both the sublimation front as well as the microstructure in more detail.

While 3D imaging gives detailed information on the product and the microstructure, the imaging process typically takes quite long and it also requires sophisticated equipment, compared to 2D imaging. Because of that fact, most laboratories and R&D centers are equipped with lyomicroscopes to study freeze-drying. So, samples can be analyzed in minutes up to a couple of hours depending on the process conditions [2]. The lyomicroscope was originally developed for analyzing the collapse temperature of the product. More recently, the device has also been used to analyze the freeze-drying process in situ in 2D.

Raman (2015) [21] investigated the freeze-drying of coffee solutions. They estimated mass transfer resistance by the speed of the sublimation front. As a result, they found a strong impact of solid concentration on mass transfer resistance.

While substantial research has been carried out on microstructures with a focus on pore size distribution, other factors such as pore geometry or orientation have been scarcely studied. In a previous study, Foerst et al. [3] calculated tortuosity factors for different freeze-dried maltodextrin samples but also mainly focused on the pore size distribution to explain the findings.

In another study, Lammens et al. [22] investigated the effect of spin freezing on pore size and tortuosity. Here, they showed that a slow spin freezing rate results in a high tortuosity with a high mass transfer resistance in the dried product, while a fast spin freezing rate results in low tortuosity and low mass transfer resistance of the dried product due to a lamellar structure.

In a another study, Ray et al. [23] investigated the velocity of the sublimation front of lactose solutions by using different freezing parameters as well as different drying temperatures and pressures. They observed not only a strong impact of the pore size, but also the pore orientation with regard to the sublimation front direction.

The literature shows the significance of the pore structure on drying kinetics. While in the past only different microstructural properties have been investigated, the combined influence of pore size, shape and orientation has not been studied. Therefore, the aim of this study is to investigate the combined effect of these structural attributes in 2D on the drying kinetics with a lyomicroscope. Maltodextrin and sucrose are used as the model substance.

## 2. Materials and Methods

### 2.1. Sample Preperation

The model substances maltodextrin (Glucidex 12D, Roquette, France) and sucrose (Südzucker, Germany), both in an amorphous state, were used. Solutions with a solid concentration of c_1_ = 0.05 w/w and c_2_ = 0.15 w/w of maltodextrin and sucrose were prepared by mixing them with demineralized water. To ensure full solution, the sample solutions were gently stirred by a magnet stirrer at 300 rpm for 1 h at ambient temperature. Afterwards, the samples were stored inside a fridge at 4 °C until further use.

### 2.2. Freeze-Drying Microscope

#### 2.2.1. Setup

All freeze-drying experiments took place in the freeze-drying microscope (FDM) FDCS196 (Linkam Instruments, Tadworth, UK), which was connected to the Olympus BX51 polarized light microscope (Olympus Microscopy, Essex, UK). The freeze-drying chamber was equipped with a pirani gauge sensor and the vacuum was supplied by an Edwards E2M1.5 vacuum pump (Edwards, Burgess Hill, England). The stage was controlled by the software Lynksys32 (Linkam Scientific Instruments Ltd., Tadworth, England). To observe the freeze-drying process and the microstructure of the product, the microscope was equipped with the Olympus XC50 camera (Olympus K.K., Tokyo, Japan). The setup is shown in Figure 1.

For the experiments, 3.6 µL of the maltodextrin or sucrose solution were placed onto a 15 mm diameter quartz glass crucible, which was then inserted on a silver block inside the stage. Beneath the crucible, a small amount of silicon oil was placed for a better heat transfer. The sample was enclosed by a spacer in the form of a circular arc to achieve a uniform thickness of 330 µm, resulting in a disk of a diameter of 3.5 mm and a height of 330 µm. On the top, a glass coverslip was placed. Afterwards the stage was sealed and temperature and pressure were controlled.

For visualizing pore structure, images were continuously recorded by using a 20× magnification lens (LMPlan Fl 20×/0.40). The images were stored on the computer and used later for image analysis. All experiments were repeated three times.

#### 2.2.2. Freeze-Drying Experiments

After the sample was inserted in the freeze-drying stage it was closed, and all samples were cooled down to 10 °C with a rate of 10 K/min; they were held for 10 min to ensure that every sample had the same initial conditions. Afterwards the samples were cooled down to −38 °C. The cooling took place with two different rates: 1 K/min and 50 K/min. As soon as the final temperature was reached, the sample was kept under constant conditions for 30 min. In the subsequent drying step, the samples were dried at −38 °C and 10 Pa. The freeze-drying was finished as soon as all ice sublimes. The experimental procedure can be seen in Table 1.

To describe sublimation front velocities, the sublimation front is tracked over time in a field of view (FOV), which is the actual image. For that, every 60 s an image is taken in the same FOV. After the sublimation front moved out of the FOV, multiple images (minimum of three images) were generated for analyzing the microstructure in more detail. The experiment was finished as soon as all ice sublimed, which can be tracked by the sublimation front. At the end of the experiment, multiple images (at least five images at different positions in the sample) of the dried product were taken to analyze the representative microstructure of the whole sample (see Table 2). Here, at least more than 3000 pores were evaluated.

### 2.3. Image Analysis

For the image analysis, different inhouse MATLAB-codes were implemented. For calculating pore size distribution, the code of Rabbani et al. [24] was adapted. In a first step random ROIs in an image were chosen. The ROIs are chosen in a way that the different microstructures within a sample are represented. Afterwards the image was binarized by using an adaptive threshold. The threshold was set manually in a way that the structure was displayed with high agreement to the original image. In the next step, all pores were deleted at the edge of the ROI, this was carried out because some of the pores were cut, thus distorting the measured pore size distribution. Afterwards, a distance map was generated, and the pores were marked by a watershed transformation. Since every pore is labelled, the equivalent diameter (of a circle with the same area) can be calculated by counting each pixel inside a pore. The scale of a pixel was measured with the help of a scale bar. The steps of the image processing are visualized in Figure 2.

For calculating the shape of the pores, the regionprops function in MATLAB was used. Here, the ratio of the longest and shortest axes inside the pore was calculated (aspect ratio). To calculate these, an ellipsoid was placed inside the pore. For circular objects, a value of 1 is obtained, whereas for long pores or dendritic structures, a higher value is representative.

For measuring the pore orientation, the angle between the sublimation front and the longest axis in each pore was calculated. For analyzing, only absolute values were used, which means that an angle of 90° means that the pore was perpendicular to the sublimation front and a value of 0° represents horizontal pores. The calculation of pore orientation is explained in Figure 3a. It needs to be mentioned that for analyzing the overall pore size distribution and aspect ratio (see Section 3.1), at least five different images at different positions were evaluated. For describing the effect of different morphological microstructures on local drying kinetics, different ROIs were defined inside an image and evaluated and compared with the movement of the sublimation front. Since the FOV has an influence on drying kinetics, only structures inside an image can be compared, since an FOV on the outer side of the sample has different drying kinetics compared to an FOV further inside the sample due to the different mass transport resistances. Since the exact position inside the sample is unknown, it is not possible to compare different FOV.

For analyzing the movement of the sublimation front, the imtool function in MATLAB was used. The distance of how far the sublimation front moved between two images was calculated. For that the distance between the border, of which the sublimation front comes from, to the end of the sublimation front is calculated. Since every image is taken every 60 s, sublimation front velocities can be calculated based on the travelled distance over time. The mean value was then used for the actual movement of the sublimation front (see Figure 3b.)

For a better visualization on the combined influence of microstructural parameters on the drying kinetics, the results were plotted as a spiderweb diagram which was developed by Moses [25]. Here all three structural parameters (pore size, -shape, -orientation) and the velocity of the sublimation front is presented.

### 2.4. Determination of Critical Parameters

To determine the right freeze-drying temperature so that no change in microstructure happens during drying, critical temperatures, such as the glass transition temperature of the maximum freeze concentrated solution (Tg’) and the collapse temperature (Tc) needs to be measured first. For Tg’ determination, DSC measurements (Netzsch Polyma 214 (Netsch, Selb, Germany)) were used. A small amount of sample was transferred into a sealed aluminum pan. The sample was cooled to −50 °C with a rate of 10 °C/min, then held for 5 min and heated up to 30 °C with 10 °C/min. To analyze the glass transition temperature, the Proteus Analysis software (version 7.1.0, Netzsch, Selb, Germany) was used.

To determine the collapse temperature, the same setup as described in Section 2.2.1 was used. The sample was cooled down with a rate of 50 K/min and held at −38 °C for 30 min. For the maltodextrin samples, the sample was heated up to −15 °C and held again for 30 min. After this period, the pressure of 10 Pa was applied and the freeze-drying was started. After 5 min the temperature was increased with a rate of 1 K/min until collapse. The collapse was noticed as soon a loss in structure could be observed.

All critical parameters were determined three times. The arithmetic mean values are summarized in Table 2.

## 3. Results and Discussion

### 3.1. Impact of Freezing Parameters on Pore Structure

To investigate the influence of the microstructure on drying kinetics, different pore structures were generated first. In Figure 4, various microstructures are presented using different freezing parameters.

In Figure 4a–d the samples of maltodextrin are presented, whereas e–h represents the samples of sucrose. The structures (a,c,e,g) were obtained with 1K/min and (b,d,f,h) with 50K/min cooling rate. The samples (a,b) and (e,f) had a low solid concentration of c = 0.05 w/w and (c,d) and (g,h) had a solid concentration of c = 0.15 w/w. It can be seen that the microstructure of the different samples is very different. While for (a,b) the structure is quite heterogeneous the structure of (c–f) is quite homogenous. Additionally, the shapes of the pores are quite different. While for (c,d) the pores seem to be more circular for (g,h) longer pores with a dendritic structure are present. All mean pore sizes and shape factors of the different samples can be found in Table 3.

Table 3 represents the different microstructure parameters for different freezing conditions. The results match the observation in the images pretty well. While for the images (h–g) a more dendritic structure is present, we can also see a higher shape factor value.

In general, two different types of structures can be found: more spherulitic on one hand for the maltodextrin samples and on the other a more lamellar morphology, especially for the sucrose samples. Additionally, the pore size changes depending on the freezing parameter. Using a higher solid concentration or a higher freezing rate leads to smaller pores. This is already supported by various authors, since the nucleation temperature decreases due to the higher degree in supercooling [1,2]. When using a higher solid concentration, the pores for maltodextrin seem to be more circular, while for sucrose the pores are more lamellar in shape. In contrast a higher cooling rate seems to have only a small influence on pore shape. Since freezing is a complete stochastical event, exact predictions about pore size and shape are impossible; however, due to different freezing parameters, the morphological structure can be controlled. Although the effect of supercooling and, thus, the nucleation temperature is already well described, it is still, surprisingly, why sucrose and maltodextrin form different structures. Here it seems that the thermal properties of the materials play an important role. While the properties are already well investigated for sucrose solutions, thermal properties for maltodextrin solutions are not present in the literature and need to be investigated. However, since sucrose forms more lamellar structures at higher concentrations and thermal conductivity decreases [26], it is suggested that maltodextrin solutions have a higher thermal conductivity. This is also supported by the fact that depending on the temperature, water starts to form different ice crystal formations [27]. The lower the thermal conductivity of a sugar solution, the lower the degree of supercooling. Unfortunately, 15% solid concentration was the maximum concentration we could use due to the camera resolution, so we could not describe if higher maltodextrin concentration leads also to a more lamellar pore structure.

#### 3.1.1. Influence of Solid Content and Type of Sugar

As already shown in Table 3, sucrose forms larger ice crystals during freezing compared to maltodextrin (see Figure 5a). However, this is quite surprising since sucrose actually has a lower molecular weight and thus should form smaller ice crystals [28]. Since sugars are acting as cryprotectant, a higher amount of sucrose would lead to a higher degree in supercooling and, thus, should form smaller pores. Besides the pore size, it is also noticeable that maltodextrin has a lower aspect ratio compared to sucrose (see Figure 5b). Although for maltodextrin an increase in solid content leads to a lower aspect ratio and thus to more cellular pores, for sucrose it is vice versa. Here, a higher solid content leads to a more dendritic structure.

With a higher solid content, the pore size decreases (see Figure 6a). This is in agreement with theoretical considerations, where a higher solid content leads to a higher degree of supercooling and thus to smaller pores. Besides the effect on pore size, it could also be shown that a higher solid content leads to a more homogenous pore structure for maltodextrin, but not for sucrose (see Figure 6b). A possible explanation here can be due to the used matlab code. In general, this evaluation works best for round pores. For dendritic structures, it can happen that this tool sometimes faces problems due to oversegmentation.

#### 3.1.2. Influence of Cooling Rate

The effect of cooling rate is presented in Figure 7 and is also demonstrated in Table 3. For both sugars, a higher cooling rate leads to smaller ice crystals and thus smaller pores. This is due to the effect that a higher cooling rate leads to a higher degree of supercooling and thus to smaller ice crystals. [23,28,29]. Besides that, the cooling rate has nearly no effect on the shape factor of maltodextrin, only for sucrose a smaller change can be seen, but the difference is only small. Here, a possible explanation is that due to the higher cooling rate the scaling of the ice crystals is more precise [30]. Overall, it seems that cooling rate has a bigger influence on sucrose compared to maltodextrin, which can, again, be explained by the different chemical and thermal properties.

### 3.2. Freeze-Drying of Maltodextrin and Sucrose Solutions with Different Microstructures

While in the first part of the study different freezing conditions on microstructure formation were investigated, we want to present the effect of these different structures on drying kinetics. After generating different microstructures, the frozen samples were freeze-dried inside the lyomicroscope. The primary drying times were measured and the mean primary drying time is presented in Figure 8.

It can be seen that the mean primary drying times for the two different sugars are different. Sucrose has, for most conditions, longer mean primary drying times (between 3.5% and 35%) compared to the ones of maltodextrin. Additionally, it can be seen that with increasing solid content, the mean primary drying times increases for both sugar types. This is the same case for increasing the freezing rate.

While sucrose shows larger pores (see Table 3), we expected shorter primary drying times for sucrose compared to maltodextrin, since the freeze-drying conditions were the same. Surprisingly, for all cases, the primary drying time of maltodextrin was either shorter or comparable. The thermal conductivity for a sucrose solution with c = 0.05 w/w is around 0.583 W/mK and for c = 0.15 w/w 0.551 W/mK(both at 20 °C). Unfortunately, the thermal conductivity for maltodextrin is unknown and needs to be measured. However, it is expected that the thermal conductivity is higher [26]. This is also in contrast to the literature. Here, it is widely reported that larger pores lead to faster drying [2,3,28]. Since we could demonstrate that, depending on the freezing conditions, we have microstructures with different shapes, we assume that not only pore size plays an important role on drying kinetics, but pore shape seems to be important too. To describe this in more detail, the experiments were analyzed in more detail on the velocity of the sublimation front, pore size, shape and orientation. The results are presented in Figure 9, Figure 10 and Figure 11. However, it needs to be noted that only the pore structures of one single experiment each can be analyzed and compared. This is due to the nature of the lyomicroscope. By choosing a high resolution, only a very small field of view is possible. If we want to analyze other regions, the sample needs to be shifted inside the FDM. The distance of the FOV to the edge (which is open for diffusion) determines the local mass transfer resistance. While the position is important, the problem of the FDM is that the exact position of the sample cannot be measured. For that reason, only the structure of specific regions in each image is analyzed, presented and discussed. Nevertheless, the FDM also has the advantage that only a very thin layer (330 µm) of sample is presented. This prevents large temperature gradients inside the product and thus heat transfer occurs over the whole cross-section [17], which makes it possible to describe structural effects on drying kinetics just based on vapor transport limitations.

In Figure 9a, the local pore structure in the FDM is presented. In Figure 9b, the morphological parameters and sublimation front velocity are presented. Here the pore structure is quite heterogeneous and the pores are more likely like a cellular structure (lower aspect ratio) with different sizes. The mean pore size of ROI1 (9.0 µm) is smaller compared to ROI2 (11.2 µm). Additionally, a difference in the pore shape can be seen. In mean, the pores in ROI1 have a higher aspect ratio of around 17% and are thus more of a lamellar shape. The orientation of the pores (~47°) are the same for both cases. Not surprisingly, the velocity of the sublimation front for ROI2 is faster (18.1 µm/min vs 14.7 µm/min), due to larger pores. In Figure 10, a microstructure that differs in pore orientation is analyzed. 

In Figure 10 the results of the freeze-drying of maltodextrin c = 0.15 w/w is presented. Here, three different ROIs are defined. In Figure 10a, the image of the freeze-dried structure is presented, while in Figure 10b, the quantified parameters are displayed. In general, the structure of maltodextrin c = 0.15 w/w is more homogenous compared to c = 0.05 w/w. Additionally, the pores are smaller (~5 µm). This is due to the fact that a higher solid content leads to a higher rate of supercooling and thus to more ice nuclei, which results in a smaller and more homogenous structure. While for the most parts of the structure, a more cellular morphology is present, there are also regions where a more dendritic form is present. This can be also seen in the shape factors of the different ROIs. While for ROI1 and ROI3, a factor of around 2.1 is present for ROI2, a factor of around 2.6 can be measured. This can also be seen in the image since in ROI2, a more dendritic structure is present. While for ROI1 and ROI3, the orientation of the pores is also quite similar with 45°, for ROI2 an orientation of around 57° can be found. By comparing the velocity of the sublimation front, ROI3 (5.3 µm/min) is faster compared to ROI1 (4.8 µm/min), since here larger pores are present. If ROI2 and ROI3 are compared, we can see that ROI3 has a slightly larger mean pore size of around 5%; however, the velocity is a little bit slower (~2%), which is surprising. Since the sublimation front moves from the left to right, the distance to the edge and thus mass transfer resistance should be the same. However, if we take into account that for ROI2 we have a more dendritic structure and thus longer pores and the orientation of the pores, which is around 57° for ROI2, we can explain this finding. With this orientation, the pores are more perpendicular to the sublimation front and therefore allow water vapor to travel easier through the dried layer [23]. This would explain that while ROI3 still has larger pores compared to ROI2, it is still not faster in sublimation front velocity, since here the water vapor resistance is higher. Based on this, structures with a higher aspect ratio need to be investigated to prove the theory that pore orientation can be more relevant than pore size.

In Figure 11, freeze-drying of sucrose with c = 0.15 w/w is presented. Here, mostly a dendritic structure is present. In this image (Figure 11a), two ROI are defined to describe the influence of the microstructure on sublimation front velocity. ROI1 (5.29 µm) has bigger pores compared to ROI2 (4.74 µm). While in ROI1 mostly dendritic structures are present in ROI2, some cellular structures can also be seen. This is also represented by the different shape factors. For ROI1, the shape factor is around 3.8, while for ROI, the shape factor is 3.2. Again, a higher shape factor represents more longitudinal pores and thus a larger fraction of dendritic pores. If we take a closer look at the sublimation front we can see that the local velocity was faster for ROI2, even if ROI1 has larger pores, so a faster drying would be expected. As already explained for Figure 10, depending on the shape factors, it also seems that pore orientation plays a crucial role on drying kinetics. For ROI1, a mean pore orientation of 40° can be found, while for ROI2, the mean pore orientation is 49°. This also supports the findings in Figure 10. However, the influence of the pore shape and orientation is more obvious. While in the literature it is reported that larger pores lead to faster drying due to the lower water vapor resistance, we can demonstrate that depending on the shape factor the pore orientation may be more important. Here, the pores of ROI2 are more perpendicular orientated compared to the pores of ROI1 and also have more longitudinal pores, hence water vapor can more easily be transported. This is due to the reasons that less amorphous walls and thus less throats are present for the same distance compared to a cellular structure and, consequently, the water vapor resistance is lower, which results in a higher sublimation front velocity [11,23]. This observation is also supported by Figure 12a. Here, the sublimation front in a freeze-drying experiment of sucrose with c = 0.15 w/w is presented. Needles, which represent the dendritic structure, dry much faster, which are more perpendicular compared to pores which are mostly parallel to the sublimation front. Dendrites that are more perpendicular dry faster than dendrites parallel to the sublimation front. Not only local differences in sublimation front velocity can be observed, but global differences can also be demonstrated, as presented in Figure 12b. Here, regions are presented where the sublimation front already moved further compared to others and thus is due to the different microstructure of the sample. This was also reported in earlier studies where we used neutron imaging to describe the structured sublimation front. Here, a lamellar structure is also seen, which is perpendicular to the actual sublimation front, dries much fast and supports our findings [31]. Additionally, in a recent study, we could demonstrate that depending on the microstructure we have completely different movements of the sublimation front [32]. This shows, again, that it is very important to know the actual microstructure of the product and thus to control the freezing step.

## 4. Conclusions

In this study, we present the relevance of the microstructure on local drying kinetics. Here, not only the relevance of the pore size could be demonstrated, but also the relevance of pore orientation depending on the pore shape. By using the FDM, it is possible to simultaneously analyze microstructural parameters, as well as local drying kinetics. Due to the very thin layer, the whole freeze-drying process is controlled by mass transport resistance and thus the microstructure plays an important role. In the first part of this study, we analyzed the influence of different freezing parameters, such as type of sugar, solid concentration and cooling rate, on the microstructure of the whole sample after freezing drying. In general, maltodextrin forms smaller pores with a lower aspect ratio and thus more cellular/round-like pores. In contrast, sucrose has bigger pores and, especially for higher solid concentrations, pores with a higher aspect ratio, resulting in more lamellar-like structures. While the effect of cooling rate and solid concentration on pore size is already well discussed in the literature [33,34,35], the differences in aspect ratio and the influence of the type of sugar on the microstructures still needs to be analyzed. Based on the chemical and thermal properties of the different materials, we suggest that sugars with a higher thermal conductivity tend to form pores with a lower aspect ratio and vice versa. Since sucrose forms larger pores and it is widely reported that larger pores lead to faster drying, it was surprising that maltodextrin dried faster for the same drying conditions. Since both sugars form different pore structures, it seems that another factor is relevant. A typical method to increase pore size is the use of an annealing step during the freezing process. Here, the temperature is raised above Tg’ and held for a certain amount of time. During that time, bigger ice crystals are formed, which results in a shorter drying time. Next to the formation of larger ice crystals, the homogeneity between the ice crystals is also increased [3,32]. Therefore, we expect a lower shape factor. Additionally, it could be demonstrated that the connectivity increased between the pores, resulting in a lower mass transfer resistance [36].

In the second part of this study, we analyzed the influence of different local microstructures on local drying kinetics. The results indicate that not only is pore size important, but pore shape and the orientation of the pores is also significant. The influence of pore orientation on local sublimation front velocities depends on the aspect ratio of the pore. The higher the aspect ratio is, the more relevant the pore orientation is. In general, the pore orientation seems to play a role for an aspect ratio larger than 2.5. Here, pores where the orientation is most likely parallel (0°) to the sublimation front leads to slower movement, while perpendicular pores (90°) dry much faster. This is due to the fact that for the same distance, more amorphous walls and throats are present for parallel pores and thus results in a higher vapor transport resistance. For pores with an aspect ratio of below 2.5, pore orientation seems more irrelevant and, in that case, pore size is the dominating factor. This is a very important fact, because, depending on the shape of the pores, controlled freezing can have a major influence on drying kinetics and, thus, is important for operation costs.

Overall, the results are very promising and need to be evaluated further. The authors are aware of the fact that the results only represent the microstructure of the surface in 2D. For that, investigations are currently underway for freeze-drying experiments inside a micro-CT, as described in Gruber et al. [15]. By analyzing the complete microstructure in 3D, the authors expect the same outcome, since pore shape and orientation also affects tortuosity and thus mass transport resistance.

Since both maltodextrin and sucrose were in an amorphous state, it needs to be taken into account that for usual pharmaceutical applications, much more complex formulations, where also crystalline ingredients are present, are used. While the freezing behavior of amorphous and crystalline samples are different, we expect an impact, especially on the shape of the pores, which has an effect on freeze-drying kinetics. However, the influence of pore orientation is still expected to be comparable, since the influence on mass transport resistance is similar. To analyze the effect of crystalline formulations, future analyses will be conducted.

## Figures and Tables

**Figure 1 pharmaceutics-14-02132-f001:**
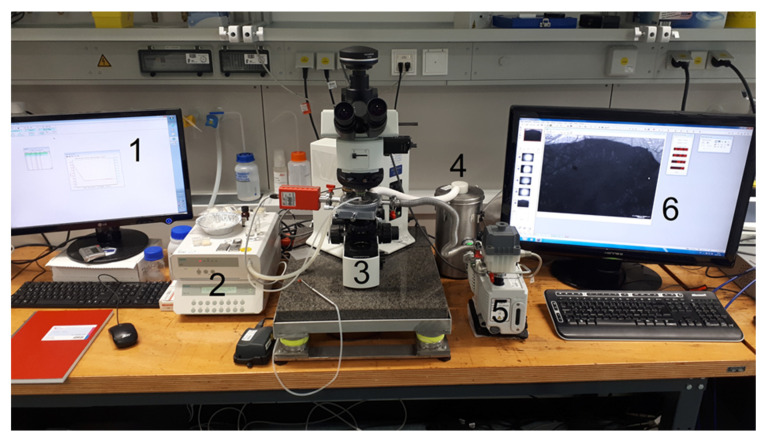
Experimental setup of the lyomicroscope: (1): Computer to controll the freeze-drying stage; (2) Liquid nitrogen pump; (3) microscope with mounted freeze-drying stage; (4) liquid nitrogen container; (5) vacuum pump; (6) computer for imaging.

**Figure 2 pharmaceutics-14-02132-f002:**
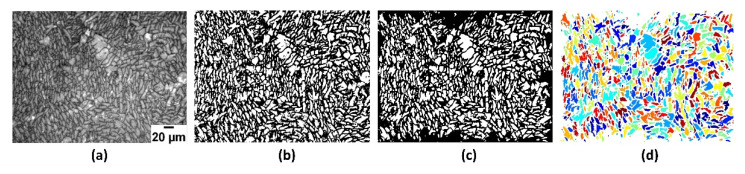
Illustration of all steps performed for image analysis for measuring pore size distribution with the inhouse MATLAB Code: (**a**) original gray scale image; (**b**) binarization; (**c**) morphological operations; (**d**) labelled watershed image.

**Figure 3 pharmaceutics-14-02132-f003:**
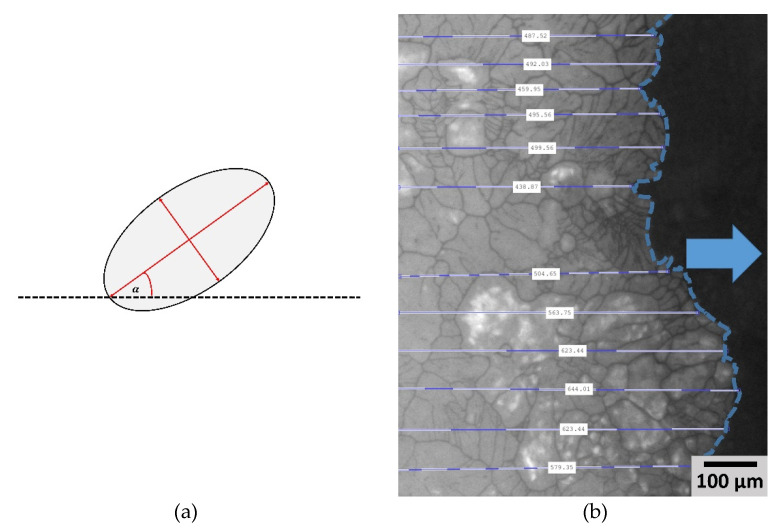
Illustration of (**a**) measuring the pore orientation and aspect ratio and (**b**) for calculating sublimation front velocity.

**Figure 4 pharmaceutics-14-02132-f004:**
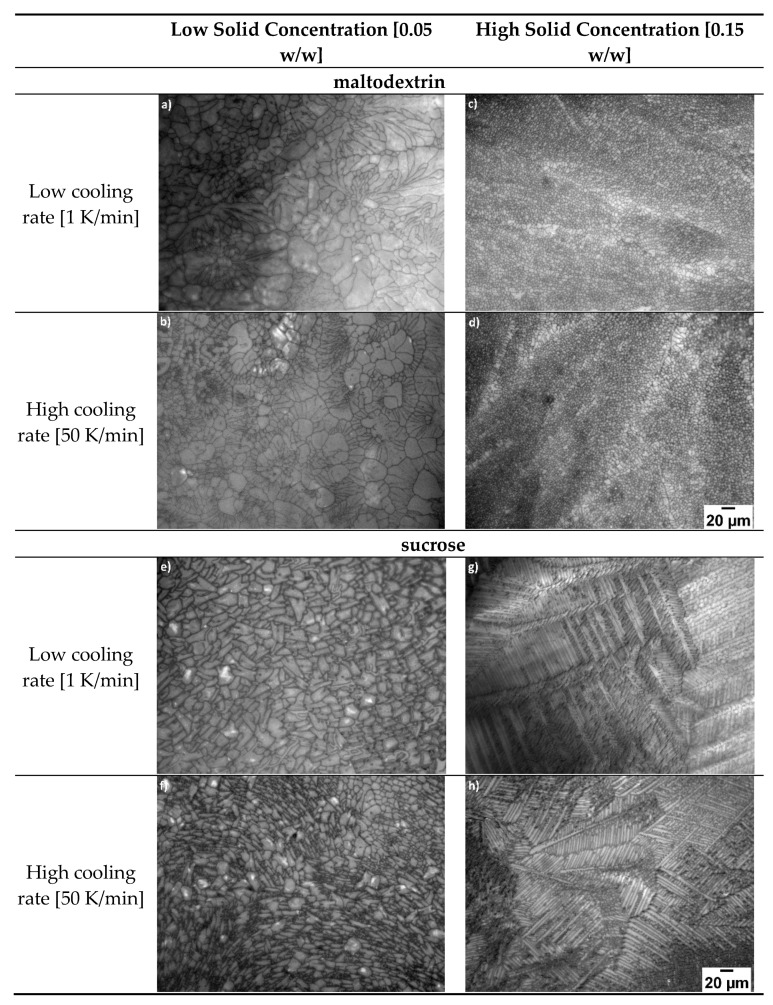
FDM images of microstructures obtained with different freezing parameters: (**a**–**d**): maltodextrin solution and (**e**–**h**): sucrose solution; (**a**,**c**,**e**,**g**) use of low (1 K/Min) cooling rate; (**b**,**d**,**f**,**h**): high (50 K/min) cooling rate; (**a**,**b**) and (**e**,**f**): low solid content (c = 0.05 w/w); (**c**,**d**) and (**g**,**h**): high solid content (c = 0.15 w/w). Scale Bar shown in (**d**,**h**) applies for all images.

**Figure 5 pharmaceutics-14-02132-f005:**
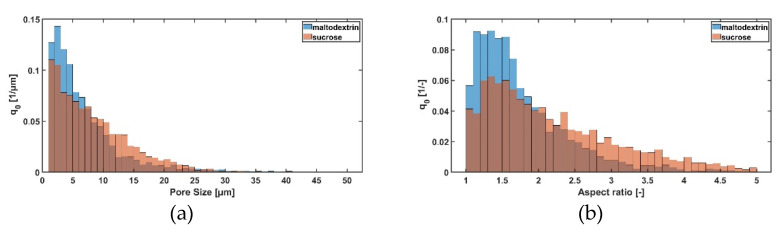
Influence of sugar type (c = 0.05 w/w; 1 K/min cooling rate) on (**a**) the pore size distribution and (**b**) aspect ratio.

**Figure 6 pharmaceutics-14-02132-f006:**
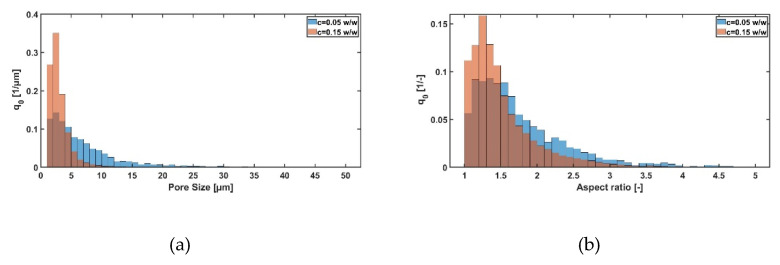
Influence of different solid concentration for a maltodextrin solution (1 K/min cooling rate) on (**a**) the pore size distribution and (**b**) aspect ratio.

**Figure 7 pharmaceutics-14-02132-f007:**
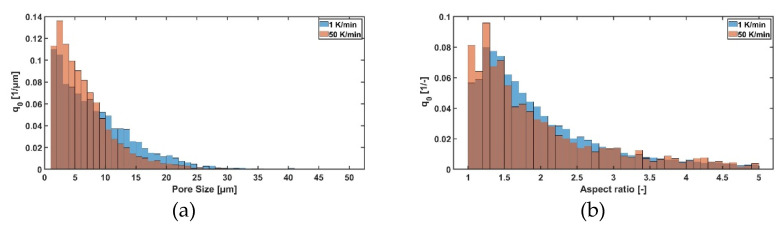
Effect of different cooling rates of a sucrose solution (c = 0.05 w/w) on (**a**) pore size distribution and (**b**) aspect ratio.

**Figure 8 pharmaceutics-14-02132-f008:**
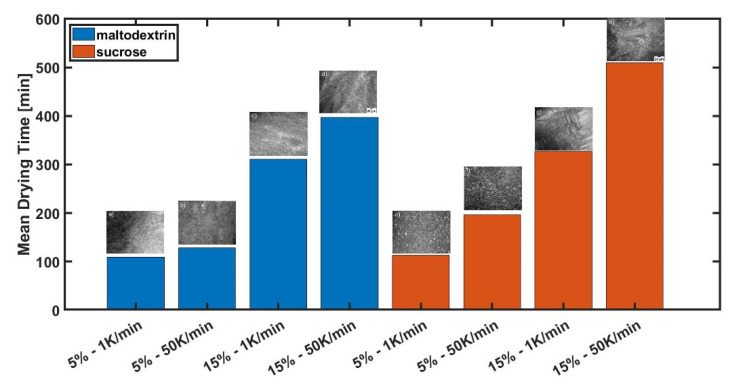
Mean primary drying times.

**Figure 9 pharmaceutics-14-02132-f009:**
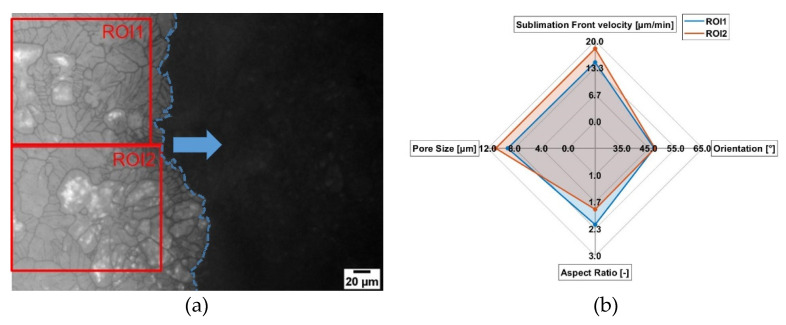
Influence of the microstructure on the local sublimation front velocity for maltodextrin (c = 0.05 w/w, 1 K/min cooling rate). (**a**) Lyomicroscope Image and (**b**) Results.

**Figure 10 pharmaceutics-14-02132-f010:**
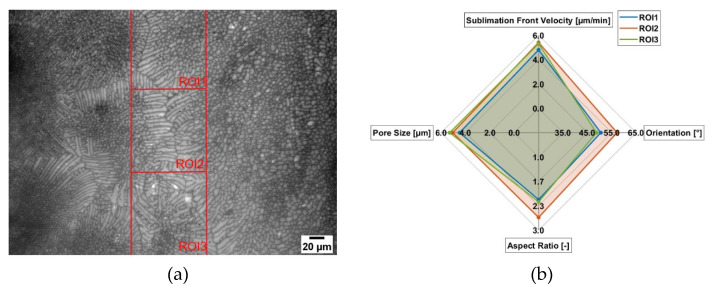
Influence of the microstructure on the local sublimation front velocity for maltodextrin (c = 0.15 w/w, 1 K/min cooling rate). (**a**) Lyomicroscope Image and (**b**) Results.

**Figure 11 pharmaceutics-14-02132-f011:**
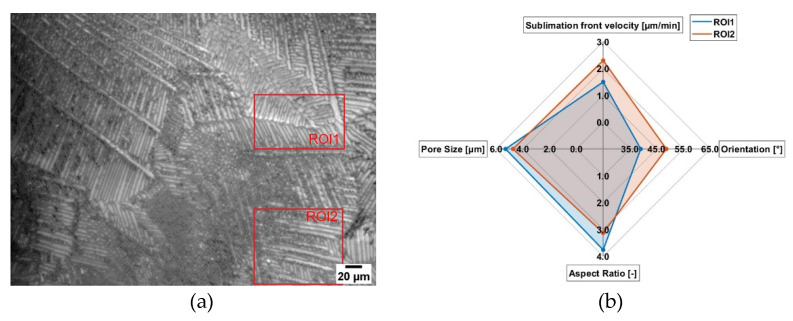
Influence of the microstructure on the local sublimation front velocity for sucrose (c = 0.15 w/w, 50 K/min cooling rate). (**a**) Lyomicroscope Image and (**b**) Results.

**Figure 12 pharmaceutics-14-02132-f012:**
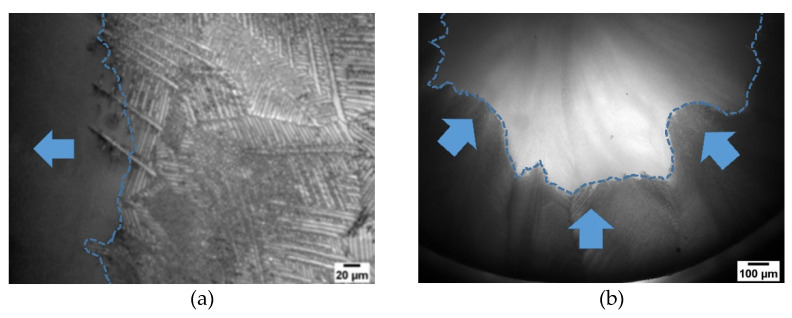
Images (Sucrose solution (c = 0.15 w/w with 50 K/min cooling rate)) of fissured sublimation fronts due to the local microstructure: (**a**) high resolution to resolve microstructure; lamella structure which is perpendicular to the drying front dries faster. (**b**) Lower resolution to resolve global sublimation front; areas which dries faster compared to others due to different microstructures.

**Table 1 pharmaceutics-14-02132-t001:** Temperature rate and limit for the freeze-drying cycle with the used holding time.

Rate [K/min]	Limit [°C]	Holding Time [min]
10	10	10
1/50	−38	30
0	−38	until experiment is finished

**Table 2 pharmaceutics-14-02132-t002:** Overview of Tc and Tg’ for maltodextrin and sucrose solutions with different solid concentrations.

	Tg’ [°C]	Tc [°C]
maltodextrin c = 0.05 w/w	−11.73	*
maltodextrin c = 0.05 w/w	−11.36	−7.86
Sucrose c = 0.05 w/w	−34.20	−29.63
Sucrose c = 0.15 w/w	−33.40	−30.86

* Couldn’t be determined.

**Table 3 pharmaceutics-14-02132-t003:** Overview of the structural parameters of different microstructures. Arithmetic mean values with confidence interval with 0.95 confidence level.

Type of Sugar	Solid Concentration [w/w]	Cooling Rate [K/min]	Pore Size [µM]	Shape Factor [-]
Maltodextrin Figure 4a–d	0.05	1	7.0 ± 0.28	1.8 ± 0.03
	0.05	50	6.8 ± 0.21	1.9 ± 0.02
	0.15	1	3.1 ± 0.03	1.5 ± 0.01
	0.15	50	3.0 ± 0.02	1.5 ± 0.00
Sucrose Figure 4e–h	0.05	1	8.4 ± 0.16	2.2 ± 0.03
	0.05	50	6.6 ± 0.14	2.1 ± 0.03
	0.15	1	4.7 ± 0.06	2.3 ± 0.03
	0.15	50	4.0 ± 0.12	2.6 ± 0.09

## Data Availability

Not applicable.

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
