# Peer review of "The Influence of Local Microstructure Inhomogeneities on Local Drying Kinetics during Freeze-Drying"

_pharmaceutics, 2022, doi:10.3390/pharmaceutics14102132_

Round 1
Reviewer 1 Report
The authors describe the impact of the microstructure of the cake on drying kinetics for two different amorphous formulations. The paper is well structured and could be of interest for the readers of Pharmaceutics. However, the study includes only one specific technology (freeze-drying microscopy). Therefore, cross-validation of the results with a second technology might be useful. Furthermore, adding other technologies might increase the novelty of the work since the impact of the microstructure on drying kinetics was already the scope of several other research papers (see citation list). In addition, only amorphous formulations were included in the study while the crystallinity might impact (alter) the conclusions. In general, I would recommend to publish the paper in Pharmaceutics after some major modifications.
Please also take following comments and suggestions into account:
· * Line 85: Wrong citation. 16= in-situ measurement of spin freeze-dried samples. Freezing rate was not part of the scope of this paper. Correct citation:
Lammens, J.; Goudarzi, N.M.; Leys, L.; Nuytten, G.; Van Bockstal, P.-J.; Vervaet, C.; Boone, M.N.; De Beer, T. Spin Freezing and Its Impact on Pore Size, Tortuosity and Solid State. Pharmaceutics 2021, 13, 2126. https://doi.org/10.3390/pharmaceutics13122126
· * Line 94: It is clear that shape and orientation of the pores impact the drying kinetics. However, analyzing the orientation of pores in a 2D image might be not that useful since the tortuosity depends on the alignment of the pores along the sample (= in 3D). Can the authors provide some additional explanation?
· * Line 102: Please include the collapse temperature of these formulations since exceeding the collapse temperature might impact the microstructure of the cake.
· * Line 102: Did the authors also consider to use crystalline formulations?
· * Line 130: Did the authors used polarized light to analyze the samples?
· * Line 148: Please add the resolution of these images (size indicator), general remark for all images.
* * Line 157: did you investigate the effect of threshold selection on the pore size/orientation? Any comment on the variability linked to threshold selection?
· * Line 162: can you motivate why this descriptor was used to calculate the diameter of the pores? Did the authors consider the use of any other pore size descriptor?
· * Please split up figure 4. Currently, the lay-out and classification of this figure is confusing and hard to interpret. A separate figure per formulation or per freezing rate might help to interpret the results.
· * Table 2, shape factor: the authors report a rather high variability for the shape factor. Please ad a statistical analysis. Please elaborate on the high variability (uncertainty) of this parameter
· * Table 2: Please adjust the significant numbers. Is it meaningful to have 4 decimal digits? Please link this to the precision of the used technology.
· * Line 287: How was the total drying time determined?
· * General remark: please try to quantity your results. For example: line 328 velocity is faster: how much faster? Or how much larger? Significant?
Author Response
Dear reviewer,
thank you for reviewing our manusscript and for your constructive and helpful comments which, in our opinion, improved our manuscript significantly. Please find attached a detailed response.

Reviewer 2 Report
The work is original and brings novelty to the field.
The advanced experimental technique allowed obtaining results with great accuracy. The interpretation of the results is clear and detailed
The quality of figures 5, 6, and 7 could be improved.
A continuation of the research could take into account the interactions between the component and water.
Author Response

(The authors gave the same response as above.)

Reviewer 3 Report
The process of freezing a formulation to form ice crystals and the maximally concentrated freeze concentrate is critical to design a lyo cycle with the optimum use of resources and time to get the desired product, which is reconstitutable and has the adequate cake appearance with drug product stability. Conventionally, formulators at the industry start with a conservative approach where they use DSC and Freeze-Drying Microscopy (FDM) to identify the formulation Tg’, Te, etc. For micro collapse events, a Differential Thermal Analyser (DTA) is used. Then, for developing a lyo cycle, some formulators adopt a DOE approach, which again is a very tedious task in comparison to the traditional trial-and-error method using the formulators' knowledge and skills. With that said, using FDM to study the cake microstructures in detail is usually not pursued until necessary. The authors have devised a promising technique to pursue that avenue. However, the applicability of the conclusion from the study might not be necessarily extended to the formulations with the commonly observed mix of amorphous and crystalline states. Moreover, the FDM with small sample sizes allows a very accurate control over freezing and drying conditions, hence properties seen in a FDM do not strictly represent what happens uniformly in vials within a lyostar laboratory scale freeze dryer or a larger commercial/production scale one. Especially when the rate of drying varies from vial to vial, which can be still brought under significant control with controlled nucleation. The influence of pore size is quite well known, but the orientation of pores is something unique and not usually considered. I have some major comments which can be easily addressed by the authors.
Major comments
1) Please provide the readers in the discussion section, information on the applicability of the conclusions (pore size and orientation) on products having amorphous (such as amorphous sucrose as a protein stabilizer) and crystalline states (eg mannitol as a bulking agent). If adequate information cannot be provided, then this needs to be discussed as a drawback of the paper, requiring exploration in future with more complex formulations.
2) The authors have used simple sugar solutions as formulations. Please, mention the state of Maltodextrin (amorphous?) and sucrose (crystalline?) in the paper.
3) Can the authors also include a brief comment on the applicability of annealing in improving the microstructural properties?
4) The method and experiments followed by the authors are important. The authors mention in-house MATLAB codes in the paper. If possible, please deposit, the MATLAB codes for image analysis so that it's available to the public. Will save a formulator considerable time and effort.
Author Response

(The authors gave the same response as above.)

Round 2
Reviewer 1 Report
I would like to congratulate the authors for the adjustments made. The paper has been significantly improved and can be published in the present form in pharmaceutics.
Reviewer 3 Report
Thank you for addressing my comments.